# OpenReview forum: "Mitigating Selection Bias with Node Pruning and Auxiliary Options"
_ICLR.cc/2025/Conference — Submitted to ICLR 2025_

### Official Review · Reviewer_fF6T · 2024-11-02

**Soundness:** 3
**Presentation:** 3
**Contribution:** 3
**Rating:** 6
**Confidence:** 4

**Summary:**

This paper addresses selection bias in large language models (LLMs) used in multiple-choice question answering (MCQ) tasks. It introduces two methods: Bias Node Pruning (BNP), which reduces bias by pruning specific parameters in the final layer of the model, and Auxiliary Option Injection (AOI), which includes an “I don’t know” (IDK) option to the choices provided. Additionally, the paper proposes Choice Kullback-Leibler Divergence (CKLD) as a new metric for measuring selection bias, aiming to improve accuracy and adaptability over existing metrics. Experimental results demonstrate the effectiveness of BNP and AOI across various datasets and LLMs.

**Strengths:**

* The paper’s parameter-level debiasing through BNP and simple input modification with AOI offer new perspectives on reducing selection bias beyond typical input/output adjustments.
* The CKLD metric is well-motivated provides good insights by accounting for class imbalance, which is important for accurate and robust evaluation.
* The extensive empirical studies justify most of the paper's claims. Also, the experiments span multiple datasets and models, demonstrating the applicability of the methods across different settings.

**Weaknesses:**

**1. Motivation 1 in Section 2.2.**

The claim that “Selection bias is amplified when the model is incorrect” does seem to misrepresent selection bias by framing it as a consequence of incorrectness rather than a pre-existing condition that affects the model’s choices. Selection bias in model predictions typically arises from the model’s predisposition toward certain options regardless of their correctness. Thus, it is more reasonable to view selection bias as a factor that can influence or even lead to incorrect responses, rather than something that is “amplified” by incorrect answers.

**2. Weak Justification for AOI.**

In Section 3.2, the hypothesis that an “I don’t know” (IDK) option would reduce selection bias due to its presence in incorrect answers is not straightforwardly justified. The logic does seem somewhat circular: if bias is inherent and present regardless of correctness, introducing an IDK option may not directly address or reduce selection bias without a clearer rationale for how it interacts with the bias mechanism. The approach could use a more robust theoretical basis to explain why AOI would specifically reduce the preference for biased options. An alternative explanation or additional empirical evidence supporting this choice would strengthen the section’s argument.

**Questions:**

1. The motivation for AOI could be strengthened by exploring additional theoretical or empirical justifications. For instance, could the authors test alternative auxiliary options, or provide analysis to show that the IDK option specifically reduces bias compared to other options?
2. BNP selectively prunes nodes, but the potential impact of this pruning on model accuracy across other types of tasks is not fully addressed. How stable is the pruning process across different datasets? Including a sensitivity analysis on how much accuracy is affected by varying pruning parameters could help further clarify BNP’s general applicability.
3. Since the model’s selection bias is observed to be prominent in the final layer, could additional experiments clarify why this layer is particularly sensitive to bias? This might provide insights that could further refine the BNP approach.

---

> ### Author Response · Authors · 2024-11-23
>
> We sincerely appreciate the reviewer's constructive comments and insightful questions. We have addressed the issues outlined below and look forward to further discussions. Please review our responses and let us know if there are any remaining concerns.
>
> >
>
> ## Q1. Motivation 1 in Section 2.2.
> The purpose of this subsection was to motivate the extraction of the "bias vector." However, we understand the potential for confusion it may cause. We have made appropriate modifications to the manuscript to address this issue. Thank you for bringing this to our attention.
>
> >
>
> ## Q2. Justification for AOI with alternative auxiliary options
>
> When collecting answers from humans, including an ''I don't know'' response can improve data quality [1]. Because the models were more likley to show selection bias when they were incorrect, we hypothesized that offering an ''I don't know'' option would improve the quality of the responses provided by the model.
>
> To support our claim, we have already included an ablation experiment in Table 4 of the manuscript to demonstrate the effect of alternative auxiliary options. Specifically, we compared the "I don't know" option with "I know the answer." Here, we additionally tested the "None of the above" option, and the results are reported below. Overall, our "I don't know" AOI consistently outperforms the alternatives across all four metrics. Notably, other option contents may degrade performance for Mistral and Bloomz. We have updated Section 6.2 of the manuscript to include the results.
>
> |  | Acc | F1 | RSD | CKLD |
> |---|:---:|:---:|:---:|:---:|
> | Llama-3 | 41.8 | 46.7 | 1.021 | 0.589 |
> | Llama-3 w/ "None of the above" | 42.4 | 42.7 | 0.833 | 0.487 |
> | Llama-3 w/ "I know the answer" | 45.6 | 46.5 | 0.790 | 0.366 |
> | **Llama-3 w/ "I don't know"** (Ours) | 48.3 | 50.5 | 0.531 | 0.288 |
>
> |  | Acc | F1 | RSD | CKLD |
> |---|:---:|:---:|:---:|:---:
> | Mistral | 46.4 | 47.6 | 0.366 | 0.186 |
> | Mistral w/ "None of the above" | 48.0 | 47.8 | 0.596 | 0.159 |
> | Mistral w/ "I know the answer" | 9.7 | 3.9 | 0.762 | 1.888 |
> | **Mistral w/ "I don't know"** (Ours) | 48.6 | 49.3 | 0.309 |0.140 |
>
> |  | Acc | F1 | RSD | CKLD |
> |---|:---:|:---:|:---:|:---:
> | Bloomz | 28.0 | 32.8 | 1.003 | 0.661 |
> | Bloomz w/ "None of the above" | 26.5 | 25.9 | 0.730 | 0.518 |
> | Bloomz w/ "I know the answer" | 28.0 | 26.1 | 0.618 | 0.314 |
> | **Bloomz w/ "I don't know"** (Ours) | 32.0 | 33.3 | 0.672 | 0.205 |
>
> [1] Converse, Jean M., and Stanley Presser. 1986. Survey Questions: Handcrafting the Standardized Questionnaire. Beverly Hills, CA: Sage.
>
> >
>
> ## Q3. Impact of BNP on different task performance
> We evaluated Llama-3's performance on two general NLP tasks—Sentiment Analysis and Text Summarization—by pruning 8, 16, and 32 nodes. For Sentiment Analysis, we used the "Multi-class Sentiment Analysis Dataset" [1], and for Text Summarization, we used the "CNN/DailyMail Dataset" [2]. The results are presented in the tables below, with the top table corresponding to Sentiment Analysis and the bottom table to Text Summarization.
>
> We observed a slight decline in performance as more nodes were pruned; however, the degradation was not severe enough to significantly affect general linguistic performance. Given that our method is specifically designed for multiple-choice question (MCQ) tasks, we believe that a minor decrease in performance on general NLP tasks is not a significant concern.
>
> | # Pruned Nodes | F1 | Acc |
> |:---:|:---:|:---:|
> | 0 | 32.7 | 22.0 |
> | 8 | 32.7 | 22.7 |
> | 16 | 31.7 | 20.2 |
> | 32 | 31.3 | 20.6 |
>
> | # Pruned Nodes | ROUGE-L | ROUGE-1 |
> |:---:|:---:|:---:|
> | 0 | 13.8  | 20.4 |
> | 8 | 13.8 | 20.2 |
> | 16 | 11.8 | 17.1 |
> | 32 | 11.5  | 16.6 |
>
> [1] https://huggingface.co/datasets/Sp1786/multiclass-sentiment-analysis-dataset
>
> [2] https://huggingface.co/datasets/abisee/cnn_dailymail
>
> >
>
> ## Q4. Why is the final layer particularly sensitive to bias?
> Figure 3(b) and Figure 8 present detailed layer-wise analyses across multiple models, consistently showing bias concentration in final layers. This aligns with established understanding that later transformer layers handle higher-level semantic tasks while earlier layers process more basic features. The effectiveness of final layer pruning (demonstrated in Tables 1-2) empirically validates our focus on this layer. We hypothesize this is because the final layer directly maps to output probabilities, making it particularly susceptible to systematic biases in token selection. Section 2.2 provides mathematical analysis showing how final layer parameters interact with bias vectors. While expanding this analysis could be interesting future work, our current results already demonstrate the practical utility of targeting this layer.

---

> ### Author Response · Authors · 2024-11-27
> **Soft Reminder**
>
> Dear reviewer fF6T,
>
> As the discussion period is coming to an end, we wanted to kindly follow up on the rebuttal responses we provided and encourage any further feedback or clarification you might have. We would greatly appreciate your thoughts on our responses to ensure we have adequately addressed your concerns.
>
> Thank you again for your time and dedication to reviewing our work. Please let us know if there are any additional points you would like us to address.
>
> Thank you.
>
> Best,
>
> the Authors.

---

> > ### Comment · Reviewer_fF6T · 2024-11-30
> >
> > Thanks for your clarification. I will maintain my positive rating and lean toward acceptance.

---

### Official Review · Reviewer_KaKJ · 2024-11-03

**Soundness:** 1
**Presentation:** 2
**Contribution:** 2
**Rating:** 3
**Confidence:** 4

**Summary:**

This paper proposes a new method to remove selection bias and introduces a new metric to quantify it. Unfortunately, the paper has fundamental methodological problems and incorrect claims. See weaknesses for more details.

**Strengths:**

With their framework, they could improve the performance of LLMs on multiple choice QA

**Weaknesses:**

- The primary problem of the paper is the experiment claiming that selection bias stems from the final decoder layer. The authors analyze the embedding differences between correct and incorrect questions within the permutation and observe a significant norm difference only in the last layer. They look at the embedding differences at each position but only report the last 50 token positions. Firstly, there cannot be any difference in the embeddings at earlier positions because LLaMA3 is a decoder-only model, meaning that the embeddings of previous tokens are not affected by future tokens. If the only difference is the order of the options, then no difference should be observed until the options are reached. It is unlikely that four options cover 50 tokens, which raises the need for an explanation from the authors regarding why a difference is still observed in early tokens in Figure 3b. Secondly, observing different embeddings at the last layer is expected because it essentially predicts the next token. How does this show bias? If the order of options is changed, and the next token after option c is analyzed, it is very likely to differ. Since the next token is different, the last layer’s embedding should differ as well. For example, the next token after the sequence “What is the capital city of France? a) Los Angeles, b) Paris, c)” and “What is the capital city of France? a) Paris, b) New York, c)” will likely differ after option c, and this difference should appear in the last layer. The authors must clarify the connection between their experimental setup and bias. The claim that “selection bias stems from the last decoder layer” is unsupported and likely untrue. It is similar to saying, “All predictions of the model stem from the decoder layer.”The last layer obviously has to be in the circuit causing this bias but this is a trivial conclusion. No experiments are needed to establish this.

- The second major issue with the paper is its proposed metrics and criticism of previously suggested metrics, namely Rstd and RSD. The authors appear confused about the definition of selection bias. They use the definition “discrepancy between the model’s response to the original choice ordering and the expected model response over all choice orderings.” However, where is the “dataset” in this definition? This definition applies to a predictive model, and a robust metric should provide similar scores across different datasets, as seen with Rstd and RSD. These two metrics are resilient to different ground truth distributions, as expected. However, the authors likely made an error in their code because while RSD appears consistent under different ground truth distributions, Rstd varies. With the proposed predictive model, recall values can be analytically calculated without needing a simulation. For option \(i\), this is equal to \(0.5 + 0.5 \times (\text{predictor's probability of sampling option } i)\). Additionally, the authors are confused about selection bias and the predictor’s performance in this experiment. For example, if a classifier randomly outputs one of the options with a 0.25 probability, according to their definition, this model would have no selection bias, even though its performance is poor. However, under the authors’ proposed Kullback-Leibler Divergence metric, the model would not receive a good selection bias score because it deviates significantly from the dataset’s ground truth distribution. Finally, the authors state that RSD gives the lowest score when the selection rate for “A” is 0.25, as if this is problematic. In reality, this is expected (Rstd would yield the same result if calculated correctly). If the model outputs uniformly at random for questions it cannot answer, it does not exhibit bias against any choice. This can be confirmed using the authors’ own definition: “discrepancy between the model response to the original choice ordering and the expected model response over all choice orderings.” This discrepancy would be 0 for such a model.

I would be happy to discuss with authors to change my score.

**Questions:**

see weaknesses.

---

> ### Author Response · Authors · 2024-11-21
>
> We sincerely appreciate the reviewer's constructive comments and insightful questions. We have addressed the issues outlined below and look forward to further discussions. Please review our responses and let us know if there are any remaining concerns.
>
>
> # 1. Clarification on the Embedding Difference Analysis in Figure 3(b)
>
> >
>
> ## 1.1 Four choices are enough to span the 50 tokens in Figure 3(b).
> We appreciate your thorough review of our preliminary analysis. To address your concern regarding soundness, we reference one sample from the ARC-Challenge dataset, which was involved in generating Figure 3(b):
> ```
> Which statement best compares single-celled and multi-celled organisms?
> (A) Tissues in a single-celled organism are like the cells in a multi-celled organism.
> (B) The nucleus in a single-celled organism is like the skin of a multi-celled organism.
> (C) Organelles in a single-celled organism are like the organs in a multi-celled organism.
> (D) The cytoplasm in a single-celled organism is like the nervous system in a multi-celled organism.
> ```
> In this sample, the Llama-3 tokenizer tokenizes the four answer choices into 92 tokens. Such samples with lengthy answer choices contribute to the difference in the early token locations of Figure 3(b). We also conducted a sanity check on the code and confirmed that **no error exists in our analysis in Figure 3(b).**
>
> >
>
> ## 1.2 Clarification on the meaning of Embedding Difference
> Thank you for raising the insightful point that "final layer embeddings must differ due to different input choice orderings." While your point is valid, we would like to clarify that the key takeaway from Figure 3(b) is "**the embedding differences are more prominent in the final decoder layers than earlier layers**".
>
> In our analysis, we computed the difference between the average embeddings of "correct" and "incorrect" questions (refer to $\mathbf{b}_\mathbf{x}$ in Figure 4(a) and Equation (1)). This eliminates the semantic information of the sample itself, while  the factor contributing to the "incorrectness" remains in the difference. Additionally, by averaging across multiple samples, we further smooth out the effect of the differences in input text. Hence, we can infer that **the embedding difference reflects the choice-ordering factor that causes incorrect responses, i.e., the Selection Bias.**
>
> >
>
> ## 1.3 Further Supporting Analysis
> To further substantiate our claim that the embedding difference reflects Selection Bias, we present an additional analysis that compares the average intra-difference within the "correct" question set and the average inter-difference between the "correct" and "incorrect" question sets. Specifically, for each dataset, we compute:
>
> $$
> \mathbf{d}_\text{intra} = \frac{1}{|\mathcal{Z}\_+ | \times |\mathcal{Z}\_+|} \sum\_{\mathbf{z}\_+^i \in \mathcal{Z}\_+} \sum\_{\mathbf{z}\_+^j \in \mathcal{Z}\_+} ||\mathbf{z}\_+^i - \mathbf{z}\_+^j||_2^2
> $$
>
> $$
> \mathbf{d}_\text{inter} = \frac{1}{|\mathcal{Z}\_- | \times |\mathcal{Z}\_+|}\sum\_{\mathbf{z}\_-^i \in \mathcal{Z}\_-} \sum\_{\mathbf{z}\_+^j \in \mathcal{Z}\_+} ||\mathbf{z}\_-^i - \mathbf{z}\_+^j||_2^2,
> $$
>
> where $\mathcal{Z}\_+, \mathcal{Z}\_-$ are the "correct" and "incorrect" embedding sets, respectively.
>
> If the embedding difference does NOT reflect Selection Bias, the average intra-difference within the "correct" embeddings ($\mathbf{d}\_\text{intra}$) should be comparable to the average inter-difference between "correct" and "incorrect" embeddings ($\mathbf{d}\_\text{inter}$). However, as shown in the table below, $\mathbf{d}\_\text{inter}$ consistently exhibits higher values than $\mathbf{d}\_\text{intra}$. This observation suggests that the embedding difference captures information correlated with the "incorrectness" of certain choice orderings, thereby reflecting Selection Bias.
> ||$\mathbf{d}\_\text{intra}$ | $\mathbf{d}\_\text{inter}$ |
> |---|:---:|:---:|
> | ARC-Challenge | 18.97 | 21.20 |
> | MMLU-Redux | 20.10 | 23.17 |
> | CommonsenseQA | 25.82 | 26.55 |

---

> ### Author Response · Authors · 2024-11-21
>
> # 2. Clarification on the definition of Selection Bias and its relation to metrics.
> With all due respect, we believe the reviewer may have interpreted the definition of Selection Bias differently. Here, we will clarify its definition and address all the questions raised by the reviewer.
>
> >
>
> ## 2.1 Where is the "dataset" in the definition of Selection Bias?
> In all previous works, Selection Bias has been discussed **in the context of specific datasets** [1][2]. In our definition, we also defined Selection Bias as the discrepancy between the model's selection under the original choice ordering (of the dataset) and the expected (average) model selection over all possible choice orderings (of the dataset). Consequently, the Selection Bias metrics—RSD, RStd, and our proposed CKLD—are inherently conditioned on the dataset being evaluated. This framing is natural, as it accounts for the diversity in question difficulty and format across different datasets. To provide clearer definitions, we have revised Section 2.1 of the manuscript and highlighted the modifications in blue. Thank you for bringing this to our attention.
>
> [1] Chunjie Zheng et al, "Large Language Models are Not Robust Multiple Choice Selectors", ICLR 2024
>
> [2] Sheng-Lun Wei et al, "Unveiling Selection Biases: Exploring Order and Token Sensitivity in Large Language Models", ACL 2024
>
> >
>
> ## 2.2 Addressing the questions
>
> ### Q1: **"a robust metric should provide similar scores across different datasets, as seen with RStd and RSD"**
>
> Datasets vary in difficulty and question format, both of which impact Selection Bias. For example, a model may exhibit lower Selection Bias on easier datasets because its confidence in selecting the correct answer outweighs its preference to specific choice options. Thus, **Selection Bias metrics should yield different scores for different datasets**. Furthermore, contrary to the reviewer's claim, the RSD values reported in Table 1 of the manuscript differ significantly across the three datasets. Similarly, as shown in an ICLR 2024 Spotlight paper [1], gpt-3.5-turbo's RStd values for ARC-Challenge, MMLU, and CSQA were 3.3, 5.5, and 2.2, respectively. These results further underscore the variability of Selection Bias metrics across datasets.
>
> [1] Chunjie Zheng et al, "Large Language Models are Not Robust Multiple Choice Selectors", ICLR 2024
>
> ### Q2: **"the authors likely made an error in their code because while RSD appears consistent under different ground truth distributions, Rstd varies"**
>
> It is unlikely that we made an error in the RStd code as the core part is taken from the code repository of [1] ([code repo](https://github.com/chujiezheng/LLM-MCQ-Bias/blob/e78748b673346f3b307e728e68eb48a0f1baf2bc/code/debias_pride.py#L153)). We reveal our RStd and RSD code below.
> ```
> CHOICES = "ABCDEFGH"
>
> def rstd(preds, labels):
>     report = classification_report(preds, labels, output_dict=True, zero_division=0)
>     recalls = []
>     for choice in CHOICES :
>         if choice in report.keys():
>             recalls.append(report[choice]['recall'] * 100)
>     return np.round(np.std(recalls), 4)
>
> def rsd(preds, labels):
>     report = classification_report(preds, labels, output_dict=True, zero_division=0)
>     accs = []
>     for choice in CHOICES :
>         if choice in report.keys():
>             choice_corr = [1 if pred == label and label == choice else 0 for pred, label in zip(preds, labels)]
>             choice_support = [1 if label == choice else 0 for label in labels]
>             acc_choice = sum(choice_corr) / sum(choice_support) if sum(choice_support) != 0 else 0.0
>             accs.append(acc_choice)
>     avg_acc = sum(accs)/len(accs)
>     acc_var = [(x-avg_acc)**2 for x in accs]
>     rsd_acc = np.sqrt(np.mean(acc_var)) / acc if acc != 0 else -1
>     return np.round(rsd_acc, 4)
> ```
> [1] Chunjie Zheng et al, "Large Language Models are Not Robust Multiple Choice Selectors", ICLR 2024
>
> ### Q3: **"if a classifier randomly outputs one of the options with a 0.25 probability, under CKLD, the model would not receive a good selection bias score because it deviates significantly from the dataset’s ground truth distribution."**
>
> Yes, that is correct! CKLD measures the divergence between the choice and the answer distribution but does not factor in the model's performance. This is why we emphasized in lines 304–305 that "it is important to refer to multiple metrics for a robust assessment." We believe that performance-based metrics (e.g., RSD) and distribution-based metrics (e.g., CKLD) are complementary and should be used together to ensure comprehensive evaluation.
>
>
> ### Q4: **"The authors state that RSD gives the lowest score when the selection rate for 'A' is 0.25, as if this is problematic."**
>
> The problem of RSD is that it gives the lowest score **regardless of the answer choice distribution in the dataset**. Ideally, the distribution of the unbiased prediction should roughly be similar to the dataset's answer choice distribution.

---

> ### Comment · Reviewer_KaKJ · 2024-11-21
>
> 1. May you report how many samples are able to cover 50 tokens with 4 options and how many of them not?
> 2. I am not saying there is no difference on the final decoder layer or similar to this. Of course, the bias problem may be reflected in the final decoder layer but this is a trivial claim. Every difference in the earlier layers may exist in the final layers because of the residual streams. It doesn't show the final decoder layer makes a real contribution. In summary: I criticize your claim which is "Selection bias stems from the final decoder layers". This is simply not supported and is very likely to be wrong.
> 3. Could you please provide the standard deviations for the mean experiment as well? This mean difference is not meaningful without stds. Besides, even though the stds are low, how do you attribute these results to selection bias? If you repeat this experiment for different layers, you may observe a similar pattern. You cannot make big claims by showing a correlation with 3 mean values. Something more convincing and possibly a causal relationship should be shown.

---

> > ### Author Response · Authors · 2024-11-25
> >
> > ## Q1.1
> > Thank you for the follow-up question. For Figure 3 (b), 31.3% of the samples had answer choices spanning over 50 tokens. We believe this is enough to highlight the earlier token locations in the figure.
> >
> > >
> >
> > ## Q1.2
> > We understand your concern regarding the potential overstatement of pinpointing the final decoder layers as the source of selection bias. In response to your comments, we have revised our claim in Section 2.2 to state that "Selection bias is prominently observed in the final decoder layers." This way, we are not overstating that the final decoder layers are the origin of the bias, but rather emphasizing that the bias is more readily observed in these layers compared to earlier ones.
> >
> > Additionally, we would like to note that this point serves as an intermediate analysis to motivate the design of our Bias Node Pruning method. As such, this change in the manuscript does not affect the overall conclusions of the paper. Thank you for your insightful and careful review.
> >
> > >
> >
> > ## Q1.3
> > We here provide the t-test results for the mean values and also conducted the same experiment on the median and first layers. As the layer moves closer to the input layer, the scale of differences decreases, and the t-test p-values increase. This indicates that the difference between $\mathbf{d}\_\text{intra}$ and $\mathbf{d}\_\text{inter}$ is more statistically significant in the final layer, suggesting that selection bias is captured more towards the last decoder layers.
> >
> > | Final Layer |  $\mathbf{d}_\text{intra}$ |  $\mathbf{d}_\text{inter}$ | t-test p-value |
> > |---|:---:|:---:|:---:|
> > | ARC-Challenge | 18.97 | 21.20 | 0.005 ** |
> > | MMLU-Redux | 20.10 | 23.17 | 0.016 ** |
> > | CommonsenseQA | 25.82 | 26.55 | 0.040 * |
> >
> >
> > | Median Layer |  $\mathbf{d}_\text{intra}$ | $\mathbf{d}_\text{inter}$ | t-test p-value |
> > |---|:---:|:---:|:---:|
> > | ARC-Challenge | 0.40 | 0.46 | 0.095 |
> > | MMLU-Redux |0.56 | 0.64 | 0.195 |
> > | CommonsenseQA | 0.43 | 0.47 | 0.204 |
> >
> >
> > | First Layer |  $\mathbf{d}_\text{intra}$ |  $\mathbf{d}_\text{inter}$ | t-test p-value |
> > |---|:---:|:---:|:---:|
> > | ARC-Challenge | 0.0054 | 0.0059 | 0.424 |
> > | MMLU-Redux | 0.0080 | 0.0093 | 0.298 |
> > | CommonsenseQA | 0.0047 | 0.0049 | 0.400 |

---

> ### Comment · Reviewer_KaKJ · 2024-11-21
>
> 2.1 Yes, the model selection bias will be evaluated in a dataset but the selection bias is something the model has or not. It's a model's feature. The dataset here is only for estimating the model's selection bias performance. In some datasets model can perform higher bias, in other datasets it can be lower. If you were able to test a model's selection bias on all possible questions in the universe, then you would get the actual selection bias. The definition shouldn't be a function of dataset D. It should be an expectation over all possible datasets.
>
> 2.2 "a robust metric should provide similar scores across different datasets, as seen with RStd and RSD". This claim is TRUE for your proposed synthetic models because that model's bias performance doesn't change from one dataset to another dataset. It will always make some portion correct and the other portion does randomly. However, for real models, the selection bias can be different from one dataset to another because the model for instance can have more selection bias in medical questions than mathematical questions. That's why these two metrics can output different values in different datasets. However, this difference should reflect the model's actual selection bias and should be independent of the choice distribution of the dataset.
>
> 2.3  "the authors likely made an error in their code because while RSD appears consistent under different ground truth distributions, Rstd varies". You don't need to run ANY experiments to get RSTD results with the proposed synthetic classifier. You can calculate it analytically. The recall of option (i) is 0.5 + 0.5 x  (predictor's probability of sampling option i). The first element 0.5 comes from the fact that the model predicts 50% of questions correctly no matter what and the second element comes from the probability that the model would predict correctly (by chance) in the questions whose answer is option i.  However, your plots show that, for a given classifier, RSTD outputs differently in different dataset distributions. Therefore, there has to be a problem in the code because as you see above the recall of option (i) is not a function of the choice distribution of a dataset. Lastly, you only give the code for calculating the std of recalls but it is likely that how to calculate recall is wrong (still don't need to run a simulation for that).
>
> 2.4) If a classifier outputs uniformly random then there is no selection bias. This also follows your definition. For any given option order, the model's probability to choose any content is the SAME therefore there is no bias. Mathematically, $ E[\text{output o}] = E[\text{output o} | \text{any choice order of outputs}] =  1/(\text{total possible outputs})$
>
> 2.5) "Ideally, the distribution of the unbiased prediction should roughly be similar to the dataset's answer choice distribution." This is ideal for a good performant classifier, not a classifier that doesn't have selection bias. You make the same mistake. Please, do not confuse the selection bias and the model performance. There can be a dummy model that would give terrible performance without having a bias to any option.
>
> I agree that "performance-based metrics (e.g., RSD) and distribution-based metrics (e.g., CKLD) are complementary and should be used together to ensure comprehensive evaluation" but your proposed metric cannot replace RSD or RSTD.

---

> ### Author Response · Authors · 2024-11-25
>
> ## Q2.1
> Yes, we understand your concern. However, as you may recognize, it is impractical to evaluate a model's Selection Bias across the entire data distribution. Hence, all prior studies on Selection Bias have used dataset performance as a proxy for estimating the level of Selection Bias exhibited by a model.
>
> >
>
> ## Q2.2~2.3
> To our understanding, **the recall for option (i) is NOT "0.5 + 0.5 x (predictor's probability of sampling option i)"**. Rather, it is  $0.5 + 0.5 \times P(\hat{y}= i | y = i)$. The last probability function, $P(\hat{y}= i | y = i)$, is conditioned on the ground truth label, indicating that recall inherently depends on the dataset choice distribution.
>
> For the recall function in our implementation, we use the [scikit-learn package](https://scikit-learn.org/0.15/modules/generated/sklearn.metrics.classification_report.html), which is identical to the codebase used in the ICLR 2024 Spotlight paper [1]. Also, we have re-verified the evaluation code to ensure it does not contain any errors.
>
> Nonetheless, we greatly appreciate your insightful comment regarding the importance of evaluation metrics being agnostic to dataset distribution to isolate the pure effect of Selection Bias. However, our findings suggest that none of the currently available metrics fully achieve this goal (RStd relies on recall, RSD relies on accuracy etc.). We hope this encourages further research to develop metrics that can address this limitation.
>
> [1] Chunjie Zheng et al, "Large Language Models are Not Robust Multiple Choice Selectors", ICLR 2024
>
> >
>
> ## Q2.4~2.5
> As stated in lines 261–262 of the manuscript, the experiments in Figure 5 are designed to demonstrate the impact of different data characteristics (i.e., choice distributions) when evaluated using each metric. Hence, we would like to clarify that the synthetic scenario—where half of the predictions are correct and the other half are randomly decided—**does not assume a specific model**. Rather, the synthetic predictions $\hat{Y}$ are considered to be generated by an arbitrary model with an unknown level of selection bias.
>
> And yes, we did not claim that CKLD should replace RSD or RStd. As stated in lines 304–305 and outlined in Appendix A.4, our intention is to add an additional dimension to the evaluation of Selection Bias metrics.

---

> > ### Comment · Reviewer_KaKJ · 2024-11-25
> >
> > Recall of option i is basically how much portion of i is predicted correctly when the ground truth is i. So, in the probability notation, it is $ P(\hat{y} = i |  y = i )$. Now, let's apply the law of total probability. The model outputs directly correct or outputs randomly. $ P(\hat{y} = i |  y = i ) = P(\hat{y} = i |  \text{output correctly}, y = i )  P (\text{output correctly}) + P(\hat{y} = i |  \text{output randomly}, y = i )  P (\text{output randomly})$. So, $P(\hat{y} = i |  y = i ) = 1 \times 0.5 +  P(\hat{y} = i |  \text{output randomly}, y = i )  \times 0.5$ As when the model outputs randomly, it outputs each option with some probability regardless of the ground truth $ P(\hat{y} = i |  \text{output randomly}, y = i )  = P(\hat{y} = i |  \text{output randomly})$. So, $P(\hat{y} = i |  y = i ) = 0.5 + 0.5 \times P(\hat{y} = i |  \text{output randomly})$. I hope this is clear now.
> >
> > I couldn't understand how your response addresses my concerns in 2.4 and 2.5. I basically claim that a metric that claims to measure the selection bias should output the lowest possible score for the dummy classifier I proposed.

---

> > > ### Comment · Reviewer_KaKJ · 2024-11-25
> > >
> > > I believe this paper has a good potential but the narrative should be modified and the claims should be re-examined again. I appreciate the authors's responses and additional experiments but with the current form, I cannot accept this paper.

---

> > > > ### Author Response · Authors · 2024-11-27
> > > > **Concluding Response**
> > > >
> > > > Thank you so much for the insightful and in-depth discussion. We greatly appreciate your time and commitment to reviewing our paper.  As the discussion period is coming to an end, we would like to summarize our discussion and finally attempt to address your remaining concerns.
> > > >
> > > > >
> > > >
> > > > ## Reviewer Comments
> > > > 1. The reviewer raised concerns regarding the experimental setup in Section 2.2 (Figure 7(b)):
> > > >
> > > >     (1) Questioning the correctness of the code.
> > > >
> > > >     (2) Expressing skepticism about whether the embedding difference represents Selection Bias.
> > > >
> > > >     (3) Criticizing the claim that Selection Bias originates from the final decoder layers.
> > > >
> > > > 2. The reviewer also raised concerns about the definition of Selection Bias and its relationship to the metrics (RStd, RSD, and CKLD):
> > > >
> > > >     (1) Questioning the definition of Selection Bias and its connection to the evaluation metrics.
> > > >
> > > >     (2) Questioning the validity of the experimental results in Figure 5 due to concerns about the correctness of the metric implementation.
> > > >
> > > > ## Author Responses
> > > > 1. Regarding Section 2.2 and Figure 7(b):
> > > >
> > > >     (1) To address the reviewer's concern, we have provided the ratio of samples and an actual data example to demonstrate that Figure 7(b) is not flawed and that the code functions correctly.
> > > >
> > > >     (2) We provided additional explanation clarifying the relationship between the embedding differences (i.e., the bias vector) and Selection Bias. We also included an alternative analysis showing that the embedding differences between "correct" and "incorrect" samples contain information related to Selection Bias. However, we wish to emphasize that we do not claim these embedding differences are a pure representation of Selection Bias. Rather, we would like to state that the magnitude of these differences correlates with Selection Bias information.
> > > >
> > > >     (3) We acknowledged the potential for confusion caused by our phrasing and have revised the manuscript to avoid overstating our findings. Specifically, we clarify that we do not claim that the final decoder layers are the source of Selection Bias. Instead, our intention was to show that Selection Bias is more readily observed in the final layers, which motivated our design choice to focus on output head parameters when debiasing the model.
> > > >
> > > > 2. Regarding Selection Bias definition and metric experiments :
> > > >
> > > >     (1) We recognize that Selection Bias is an inherent model property and, ideally, should not vary across datasets. However, we clarified that current evaluation metrics and datasets are limited in their ability to measure the pure level of Selection Bias intrinsic to the model.
> > > >
> > > >     (2) To address concerns about the validity of our Figure 5 analyses, we have disclosed our evaluation metric code, which is based on the codebase from prior works. This ensures transparency and confirms the correctness of our implementation. We also provided explanations on why our experimental results are not incorrect.
> > > >
> > > > >
> > > >
> > > > ## Reviewer's Remaining Concerns
> > > >
> > > > We believe the reviewer's remaining concern is mostly relevant to 2-(2). The reviewer's main point is that the experimental setting in Figure 5 is evaluating the model on a bias-free model, and if so, the minima should be consistent across "A" Label Ratios.
> > > >
> > > > >
> > > >
> > > > ## Author's Response to the Remaining Concerns
> > > >
> > > > We believe there is a fundamental mismatch in our understanding regarding the experimental setting of Figure 5. While the reviewer interprets the prediction generating mechanism (half predicting correctly and half predicting based on the "A" Selection Rate) as representing an actual model, we are instead demonstrating a hypothetical scenario where an arbitrary model rendered a certain prediction with difference preferences for choice option "A." Importantly, **the fact that we randomly generated predictions does not imply that the model itself is actually predicting randomly**. The purpose of this experiment is to simply compare how the metric trends when a particular prediction distribution is evaluated against a specific label distribution. That is, even the same prediction could have been generated from two different models with different levels of Selection Bias.
> > > > Most importantly, we would like to emphasize that this discussion is not the essence of our work. The primary focus of our research lies in the proposed methods—Bias Node Pruning and Auxiliary Option Injection—which are evaluated using RSD and CKLD.
> > > >
> > > > Additionally, if concerns regarding the implementation of RStd yet remain:
> > > >
> > > > - **We must emphasize that the evaluation code for RStd is taken from the ICLR 2024 Spotlight paper codebase.**
> > > >
> > > > - **RStd is not used in our main experiments, meaning the correctness of the RStd implementation does not affect the validity of our core results of the paper.**
> > > >
> > > > We hope these clarifications resolve your remaining concerns.
> > > >
> > > > Again, we sincerely appreciate the reviewer for the insights and comments on the manuscript.
> > > > Thank you very much.
> > > >
> > > > Best wishes,
> > > >
> > > > the Authors.

---

> > > > > ### Comment · Reviewer_KaKJ · 2024-11-27
> > > > >
> > > > > Thank you for the explanation but I know the actual model itself wouldn't generate randomly and it's a hypothetical scenario. What I showed you is that under this hypothetical scenario, you can analytically calculate RSTD with the formula I explained in the previous comment. The code you provided me was calculating std of recalls for given recalls but the problem is probably the calculation of recalls. Anyhow, still, it doesn't matter. Section 4 should be completely rewritten or omitted with this level of error.
> > > > >
> > > > > Regarding the other concerns, I appreciate the authors' effort for additional experiments and modifying the paper.  I will increase my score to 3.  However, I strongly think, the paper should be revised again and re-written. I believe, then, the paper would have a potential for acceptance.

---

### Official Review · Reviewer_zisy · 2024-11-03

**Soundness:** 2
**Presentation:** 3
**Contribution:** 2
**Rating:** 5
**Confidence:** 3

**Summary:**

This work investigates the selection bias of LLMs for multiple-choice questions. They identify two sources of selection bias: (1) LLMs tend to be more biased when they make incorrect predictions; (2) Upon examining the parameter space, the selection bias mainly arises from the last decoder layer. Based on these two observations, they proposed (1) BNP to debias LLMs by zeroing out the rows of final embedding projection that signifies the pre-computed bias vector; and (2) Adding an auxiliary option. The experimental result shows that the proposed techniques are effective across three MCQ datasets.

**Strengths:**

* This paper is well written.
* The selection bias problem of LLMs is well-motivated.
* The proposed techniques could effectively mitigate the selection bias on MCQs on the considered models and tasks.

**Weaknesses:**

* To me, Section 2 is lacking some details and needs more discussions and clarification. For example, what is the specific setting used to produce Figure 2? Is it evaluated based on zero-shot or in-context learning (or does this matter in terms of selection bias)? Are there any scaling trends since some works suggest that smaller LMs struggled to answer MCQs with the correct format under zero-shot, which might have different behavior on selection bias? Are the open-weight LMs and black-box LMs evaluated using the same criterion? (based on the probability of choice token or based on the similarity between choice option and response?) If not, does the evaluation criterion matter? I think the current evaluation is too brief to draw a holistic conclusion.

* The scope of the proposed BNP method is limited to open-weight models with choice token-based evaluation. This could possibly be addressed by demonstrating its effectiveness on the black-box settings with open-weight models, i.e., similarity-based evaluation.

* The motivation and rationale for introducing an auxiliary option to mitigate the selection bias when the model is incorrect is unclear and should be explained more than currently provided. It would also be helpful to demonstrate how the auxiliary option affects the distribution of the choices, especially for the incorrect questions.

* In Appendix A.2, line 763 -- 768, it is unclear what z['A'] + z['_A'] (especially '_A') means.

* Regarding the proposed CKLD metric, it is unclear to me why an LM needs to be able to match $p_i$, the ground truth label ratio of each specific downstream dataset, which appears to be totally agnostic to the LM. Should a uniform prior makes more sense in this case?

* Some details about how you evaluate the MCQ benchmarks are unclear. For example, are you using a fixed, specific permutation? Is it possible to draw any form of statistical significance regarding the improvements, such as using different permutations?

**Questions:**

* Have you tested the proposed method on the same model families at larger scales? Will the benefits still be the same?

---

> ### Author Response · Authors · 2024-11-23
>
> We sincerely appreciate the reviewer's constructive comments and insightful questions. We have addressed the issues outlined below and look forward to further discussions. Please review our responses and let us know if there are any remaining concerns.
>
> >
>
> ## Q1. Section 2 lacks details and needs clarification.
> - **"Is Figure 2 evaluated based on zero-shot or in-context learning?""**
>
> Figure 2 is derived with the zero-shot setting. We have updated the Figure 2 caption in the manuscript to make this clear. Thank you for pointing it out.
>
> - **"Are there any scaling trends?""**
>
> We extracted the "RSD / CKLD" values from Table 3 and conducted additional experiments with a larger variant of Llama-3, as summarized in the table below. Overall, larger models tend to exhibit lower Selection Bias across the three datasets. However, this trend is largely dependent on the specific dataset being evaluated.
>
> | Model | Param Size | ARC-Challenge | MMLU-Redux | CSQA |
> |---|:---:|:---:|:---:|:---:|
> | Bloomz | 7 Billion | 0.703 / 0.208 | 1.102 / 0.523 | 0.252 / 0.142 |
> | Mistral | 7 Billion | 0.140 / 0.036 | 0.216 / 0.069 | 0.155 / 0.031 |
> | Llama-3 | 8 Billion | 0.086 / 0.007 | 0.184 / 0.034 | 0.051 / 0.003 |
> | Llama-3 | 70 Billion | 0.024 / 0.002 | 0.122 / 0.019  | 0.073 / 0.003 |
> | Claude-3-Haiku | Unknown | 0.095 / 0.024 | 0.057 / 0.008 | 0.587 / 0.331 |
> | Claude-3-Sonnet | 180 Billion | 0.034 / 0.001 | 0.113 / 0.024 | 0.072 / 0.015 |
>
>
> - **"Are the open-weight LMs and black-box LMs evaluated using the same criterion? If not, does the evaluation criterion matter?""**
>
> In Figure 2, the open-weight LLMs (Llama3, Bloomz, Mistral) are evaluated based on choice token probability and black-box LLMs (Claude3-Sonnet) are evaluated based on Jaccard similarity of the outputs. While this distinction does not undermine the core findings, we acknowledge that it could influence evaluation results. We added a clarification on the selection bias evaluation criteria in the Figure 2 caption.
>
> >
>
> ## Q2. Demonstrate BNP on the black-box settings with open-weight models
> We tried applying BNP to the open-weight model parameters and evaluated the models in black-box settings. The results are presented in the tables below. The impact of BNP appears to be mixed in this setup, while showing greater effectiveness on particular datasets, such as CommonsenseQA.
>
> | CommonsenseQA | Acc | F1 | RSD | CKLD |
> |---|:---:|:---:|:---:|:---:|
> | Llama-3 | 69.9 | 69.8 | 0.051 | 0.003 |
> | Llama-3 + AOI | 71.3 | 71.2 | **0.030** | **0.003** |
> | Llama-3 + AOI + BNP | **71.4** | **71.3** | 0.053 | 0.007 |
> | Bloomz | 55.9 | 55.3 | 0.252 | 0.142 |
> | Bloomz + AOI | 59.2 | 58.2 | 0.180 | 0.105 |
> | Bloomz + AOI + BNP | **61.8** | **61.6** | **0.132** | **0.044** |
> | Mistral | 54.6 | 54.8 | 0.155 | 0.031 |
> | Mistral + AOI | 62.8 | 62.8 | **0.082** | **0.013** |
> | Mistral + AOI + BNP | **63.6** | **63.6** | 0.090 | **0.013** |
>
> | ARC-Challenge | Acc | F1 | RSD | CKLD |
> |---|:---:|:---:|:---:|:---:|
> | Llama-3 | 65.7 | 65.8 | 0.086 | 0.007 |
> | Llama-3 + AOI | **66.9** | **66.9** | **0.076** | **0.007** |
> | Llama-3 + AOI + BNP | 66.0 | 66.2 | 0.135 | 0.020 |
> | Bloomz | 41.9 | 42.6 | 0.703 | 0.208 |
> | Bloomz + AOI | 44.7 | 45.0 |**0.305** | 0.155 |
> | Bloomz + AOI + BNP | **45.6** | **45.4** | 0.513 | **0.030** |
> | Mistral | 55.2 | 55.2 | 0.140 | 0.036 |
> | Mistral + AOI | **59.0** | 59.0 | **0.117** | **0.020** |
> | Mistral + AOI + BNP | **59.0** | **59.1** | **0.117** | 0.021 |
>
> | MMLU-Redux | Acc | F1 | RSD | CKLD |
> |---|:---:|:---:|:---:|:---:|
> | Llama-3 | 51.9 | 52.2 | 0.184 | 0.034 |
> | Llama-3 + AOI | **52.6** | **53.0** | **0.177** | **0.033** |
> | Llama-3 + AOI + BNP | 51.9 | 52.5 | 0.214 | 0.050 |
> | Bloomz | 27.6 | 31.0 | 1.102 | 0.523 |
> | Bloomz + AOI | 29.4 | **31.8** | 0.972 | 0.413 |
> | Bloomz + AOI + BNP | **29.6** | 30.6 | **0.627** | **0.124** |
> | Mistral | 47.4 | 47.6 | 0.216 | 0.069 |
> | Mistral + AOI | **48.5** | **48.8** | **0.217** | 0.069 |
> | Mistral + AOI + BNP | 48.4 | 48.7 | **0.217** | **0.068** |
>
> >
>
> ## Q3. Rationale for using AOI to mitigate Selection Bias
> When collecting answers from humans, including an ''I don't know'' response can improve data quality [1]. Because the models were more likley to show selection bias when they were incorrect, we hypothesized that offering an ''I don't know'' option would improve the quality of the responses provided by the model.
>
> [1] Converse, Jean M., and Stanley Presser. 1986. Survey Questions: Handcrafting the Standardized Questionnaire. Beverly Hills, CA: Sage.
>
> >
>
> ## Q4. How AOI affects the distribution of the choices
> The distributional effect of AOI is already presented in Section 6.3, Figure 7. The dark-blue bar can be compared with the light-blue bar to see its effect. In all three dataset tasks, the choice distribution becomes closer to uniform when AOI is applied.

---

> ### Author Response · Authors · 2024-11-23
>
> ## Q5. it is unclear what $\mathbf{z}$['A'] + $\mathbf{z}$['_A'] means.
>
> '_A' is a token that represents "A" with a space in front of it, whereas 'A' is a one-character token. Since these two represent the same thing, we aggregate their logits $\mathbf{z}$ for accurate evaluation. We have updated the manuscript by including further explanations in Appendix A.2.
>
> >
>
> ## Q6. Why does an LLM need to match the ground truth label ratio?
> Consider a scenario in which an LLM exhibits a bias toward selecting option 'A'. In cases where the LLM is uncertain about the correct answer and resorts to random selection, it is more likely to choose 'A', resulting in a skewed overall choice distribution that diverges from the ground truth distribution. In contrast, an unbiased LLM would select options uniformly under uncertainty, producing a choice distribution that more closely aligns with the original ground truth distribution. Therefore, the extent to which an LLM's predictions match the ground truth distribution can serve as a proxy for measuring Selection Bias. We included this discussion in Appendix C.1 of the manuscript.
>
> >
>
> ## Q7. Statistical significance testing by permuting choices
> Thank you for the excellent idea. As suggested, we conducted a significance test by running the experiment eight times with randomly permuted choices. The mean performance values for all three datasets are presented in the tables below, with standard deviations shown in parentheses. **All values were statistically significant, with t-test p-values below 0.001.** We have also updated the manuscript by adding the results in Appendix D.1.
>
> | ARC-Challenge | Acc | F1 | RSD | CKLD |
> |---|---|---|---|---|
> | Llama-3 | 53.2 (1.3) | 55.4 (1.3) | 0.640 (0.142) | 0.485 (0.049) |
> | Llama-3 + BNP | 57.4 (1.0) | 58.0 (1.1) | 0.533 (0.145) | 0.304 (0.029) |
> | Llama-3 + AOI | 62.7 (1.0) | 63.0 (1.1) | 0.417 (0.133) | 0.201 (0.023) |
> | Llama-3 + BNP + AOI | 66.8 (1.0) | 66.6 (0.9) | 0.340 (0.140) | 0.121 (0.010) |
>
> | MMLU-Redux | Acc | F1 | RSD | CKLD |
> |---|---|---|---|---|
> | Llama-3 | 39.8 (1.6) | 44.4 (1.8) | 0.982 (0.097) | 0.673 (0.063) |
> | Llama-3 + BNP | 40.8 (1.7) | 44.8 (1.8) | 0.936 (0.100) | 0.595 (0.065) |
> | Llama-3 + AOI | 44.5 (1.8) | 47.0 (2.0) | 0.657 (0.097) | 0.384 (0.042) |
> | Llama-3 + BNP + AOI | 45.4 (1.6) | 47.5 (1.8) | 0.564 (0.018) | 0.346 (0.041) |
>
> | CommonsenseQA | Acc | F1 | RSD | CKLD |
> |---|---|---|---|---|
> | Llama-3 | 63.3 (1.1) | 64.2 (0.9) | 0.282 (0.026) | 0.106 (0.018) |
> | Llama-3 + BNP | 64.9 (1.1) | 65.2 (1.1) | 0.222 (0.012) | 0.073 (0.007) |
> | Llama-3 + AOI | 65.9 (0.9) | 66.3 (0.8) | 0.220 (0.020) | 0.069 (0.010) |
> | Llama-3 + BNP + AOI | 67.2 (0.6) | 67.2 (0.6) | 0.175 (0.011) | 0.052 (0.004) |
>
> >
>
> ## Q8. Performance on larger model families
> We also evaluated our methods on Llama-3-70B-Instruct, with the results presented in the tables below. While the model's baseline performance is already exceptionally high, we observe best performance when applying BNP and/or AOI in all three datasets.
>
> | ARC-Challenge | Acc | F1 | RSD | CKLD |
> |---|---|---|---|---|
> | Llama-3-70B | 89.6 | 89.6 | 0.024 | 0.002 |
> | Llama-3-70B + BNP | 89.7 | 89.6 | 0.024 | 0.002 |
> | Llama-3-70B + AOI | 91.0 | 91.0 | 0.010 | 0.000 |
> | Llama-3-70B + BNP + AOI | 91.4 | 91.4 | 0.016 | 0.001 |
>
> | MMLU-Redux | Acc | F1 | RSD | CKLD |
> |---|---|---|---|---|
> | Llama-3-70B | 67.0 | 67.1 | 0.122 | 0.019 |
> | Llama-3-70B + BNP | 67.4 | 67.5 | 0.110 | 0.018 |
> | Llama-3-70B + AOI | 68.1 | 68.2 | 0.090 | 0.010 |
> | Llama-3-70B + BNP + AOI | 68.3 | 68.3 | 0.077 | 0.009 |
>
> | CommonsenseQA | Acc | F1 | RSD | CKLD |
> |---|---|---|---|---|
> | Llama-3-70B | 77.8 | 77.9 | 0.073 | 0.003 |
> | Llama-3-70B + BNP | 78.8 | 78.9 | 0.107 | 0.013 |
> | Llama-3-70B + AOI | 79.4 | 79.5 | 0.062 | 0.001 |
> | Llama-3-70B + BNP + AOI | 79.8 | 79.8 | 0.082 | 0.009 |

---

> ### Author Response · Authors · 2024-11-27
> **Soft Reminder**
>
> Dear reviewer zisy,
>
> As the discussion period is coming to an end, we wanted to kindly follow up on the rebuttal responses we provided and encourage any further feedback or clarification you might have. We would greatly appreciate your thoughts on our responses to ensure we have adequately addressed your concerns.
>
> Thank you again for your time and dedication to reviewing our work. Please let us know if there are any additional points you would like us to address.
>
> Thank you.
>
> Best,
>
> the Authors.

---

> > ### Comment · Reviewer_zisy · 2024-11-27
> > **Reply**
> >
> > Thank the authors for their rebuttal and I appreciate the additional results and answers to my questions. In light of other reviews, I share the same feeling that the current manuscript needs a major revision before it is ready for acceptance. For example, for readers who are not super familiar with this specific literature, it would be natural to raise concerns regarding the evaluation metric (e.g., Q6 from me and Q2.1, 2.4, 2.5 from reviewer KaKJ), but the current presentation does not properly address them and more in-depth analysis and explanations are required than provided. Regarding the method part, the proposed auxiliary option method seems to make a marginal technical contribution, and the reason for choosing this specific method over other possible treatments is unclear. In conclusion, I still slightly lean against accepting this manuscript and will keep my score.

---

### Official Review · Reviewer_dYSz · 2024-11-06

**Soundness:** 3
**Presentation:** 3
**Contribution:** 2
**Rating:** 5
**Confidence:** 3

**Summary:**

The authors find that large language models exhibit a selection bias in question answering (QA) tasks, where the choice of options influences their outputs. To address this, they propose two orthogonal methods: one that removes linear layer parameters responsible for this behavior, and another that adds an "I don't know" option for the model to choose from. Experiments using three different model architectures across three QA datasets demonstrate improvement over baselines.

**Strengths:**

- The paper is well-written, and the methods are straightforward.
- The authors conduct experiments with different model architectures

**Weaknesses:**

- Pruning model weights can influence model behavior in unforeseen ways, especially since large language models (LLMs) are intended to be general-purpose.
- The BNP method appears unstable and offers only marginal performance improvements.
- Simply adding the "I don't know" option provides the best overall results; this is expected, given similar behavior observed in older models with the SQuAD v1 and SQuAD v2 datasets.
- The paper uses a limited number of datasets to demonstrate the method's effectiveness.
- The method addresses only one form of bias; the model may still use or learn to rely on other biases for predictions. See, for example, Mikula et al. (2024).

**Questions:**

- What is the impact of subset size on performance? Does a larger or smaller subset size affect the effectiveness of the proposed method?

---

> ### Author Response · Authors · 2024-11-23
>
> We sincerely appreciate the reviewer's constructive comments and insightful questions. We have addressed the issues outlined below and look forward to further discussions. Please review our responses and let us know if there are any remaining concerns.
>
> >
>
> ## Q1. Pruning model weights can influence model behavior in unforeseen ways.
> Thank you for your insightful point. First, we would like to note that extensive research has been conducted on LLM parameter pruning for efficient modeling or general-purpose applications [1][2]. However, **our approach involves pruning only a very small fraction of the model parameters**. For example, in the case of Llama-3, which has 8 billion parameters, we prune just 32 nodes—approximately 0.05% of the total model size.
>
> Furthermore, we evaluated Llama-3's performance on two general NLP tasks—Sentiment Analysis and Text Summarization—by pruning 8, 16, and 32 nodes. For Sentiment Analysis, we used the "Multi-class Sentiment Analysis Dataset" [3], and for Text Summarization, we used the "CNN/DailyMail Dataset" [4]. The results are presented in the tables below, with the top table corresponding to Sentiment Analysis and the bottom table to Text Summarization. We observed a slight decline in performance as more nodes were pruned; however, the degradation was not severe enough to significantly affect general linguistic performance. Given that our method is specifically designed for multiple-choice question (MCQ) tasks, we believe that a minor decrease in performance on general NLP tasks is not a significant concern.
>
> | # Pruned Nodes | F1 | Acc |
> |:---:|:---:|:---:|
> | 0 | 32.7 | 22.0 |
> | 8 | 32.7 | 22.7 |
> | 16 | 31.7 | 20.2 |
> | 32 | 31.3 | 20.6 |
>
> | # Pruned Nodes | ROUGE-L | ROUGE-1 |
> |:---:|:---:|:---:|
> | 0 | 13.8  | 20.4 |
> | 8 | 13.8 | 20.2 |
> | 16 | 11.8 | 17.1 |
> | 32 | 11.5  | 16.6 |
>
> [1] https://huggingface.co/datasets/Sp1786/multiclass-sentiment-analysis-dataset
>
> [2] https://huggingface.co/datasets/abisee/cnn_dailymail
>
> [3] Ma, Xinyin et al. "Llm-pruner: On the structural pruning of large language models." NeurIPS 2023.
>
> [4] Dong, Harry et al. "Prompt-prompted Adaptive Structured Pruning for Efficient LLM Generation." COLM 2024.
>
> >
>
> ## Q2. Marginal performance improvements
> To test the significance of the improvements, we conducted a significance test by running the experiment eight times with randomly permuted choices. The mean performance values for all three datasets are presented in the tables below, with standard deviations shown in parentheses. **All values were statistically significant, with t-test p-values below 0.001.** We have also updated the manuscript by adding the results in Appendix D.1.
>
> | ARC-Challenge | Acc | F1 | RSD | CKLD |
> |---|---|---|---|---|
> | Llama-3 | 53.2 (1.3) | 55.4 (1.3) | 0.640 (0.142) | 0.485 (0.049) |
> | Llama-3 + BNP | 57.4 (1.0) | 58.0 (1.1) | 0.533 (0.145) | 0.304 (0.029) |
> | Llama-3 + AOI | 62.7 (1.0) | 63.0 (1.1) | 0.417 (0.133) | 0.201 (0.023) |
> | Llama-3 + BNP + AOI | 66.8 (1.0) | 66.6 (0.9) | 0.340 (0.140) | 0.121 (0.010) |
>
> | MMLU-Redux | Acc | F1 | RSD | CKLD |
> |---|---|---|---|---|
> | Llama-3 | 39.8 (1.6) | 44.4 (1.8) | 0.982 (0.097) | 0.673 (0.063) |
> | Llama-3 + BNP | 40.8 (1.7) | 44.8 (1.8) | 0.936 (0.100) | 0.595 (0.065) |
> | Llama-3 + AOI | 44.5 (1.8) | 47.0 (2.0) | 0.657 (0.097) | 0.384 (0.042) |
> | Llama-3 + BNP + AOI | 45.4 (1.6) | 47.5 (1.8) | 0.564 (0.018) | 0.346 (0.041) |
>
> | CommonsenseQA | Acc | F1 | RSD | CKLD |
> |---|---|---|---|---|
> | Llama-3 | 63.3 (1.1) | 64.2 (0.9) | 0.282 (0.026) | 0.106 (0.018) |
> | Llama-3 + BNP | 64.9 (1.1) | 65.2 (1.1) | 0.222 (0.012) | 0.073 (0.007) |
> | Llama-3 + AOI | 65.9 (0.9) | 66.3 (0.8) | 0.220 (0.020) | 0.069 (0.010) |
> | Llama-3 + BNP + AOI | 67.2 (0.6) | 67.2 (0.6) | 0.175 (0.011) | 0.052 (0.004) |
>
> >
>
> ## Q3. AOI provides the best overall results; this is expected, given similar behavior observed in models with the SQuAD v1 and SQuAD v2 datasets.
>
> Thanks for your comment. It is not clear to us which specific work is being referred to here. If [1] is the paper in question, we would like to note that it emphasizes the importance of models recognizing unanswerable questions but does not specifically discuss the effect of including an "I don't know" option for multiple-choice question-answering tasks.
>
> [1] Rajpurkar, Pranav et al. "Know What You Don’t Know: Unanswerable Questions for SQuAD." ACL 2018.

---

> ### Author Response · Authors · 2024-11-23
>
> ## Q4. Limited number of datasets and the method addresses only one form of bias.
> We have conducted experiments on four different datasets: ARC-Challenge, MMLU-Redux, CommonsenseQA, and HellaSwag (Appendix). Considering that the ICLR 2024 Spotlight paper [1] reported results on only three datasets, we believe that using four datasets provides a sufficient basis to demonstrate the effectiveness and robustness of our proposed approaches.
>
> Furthermore, we acknowledge that LLMs are subject to various types of biases. However, the focus of our work is specifically on Selection Bias. Our goal is not to address other forms of bias, such as demographic or cultural biases, but rather to study and mitigate Selection Bias in the context of multiple-choice question-answering tasks. Also, the reviewer has mentioned the work of Mikula et al. (2024) without reference to a specific paper. Could you please provide which paper you are referring to?
>
> We sincerely hope this provides additional context and clarifies the scope of our study.
>
> [1] Chunjie Zheng et al, "Large Language Models are Not Robust Multiple Choice Selectors", ICLR 2024
>
> >
>
> ## Q5. Impact of data subset size on performance
> The size of the training set we use to extract the bias vectors has minimal impact on the overall performance. Also, since we are demonstrating a zero-shot inference task, the size of the test dataset does not affect the general trends observed in the results.

---

> ### Comment · Reviewer_dYSz · 2024-11-26
>
> Dear Authors,
>
> Thank you for your detailed response to my review. However, I remain sceptical about the proposed method. As mentioned in my review, pruning model weights can lead to unpredictable behavior, which you have not empirically evaluated (see, for example, The Super Weight in Large Language Models, Yu et al., 2024). Additionally, in tasks like QA, models often rely on multiple spurious correlations for answers (see for example, Think Twice: Measuring the Efficiency of Eliminating Prediction Shortcuts of Question Answering Models, Mikula et al., 2024. Addressing one does not guarantee that the model will not exploit others. Altering an LLM to fix a single "niche" issue is, therefore, not convincing. Moreover, adding an additional option to the model is a straightforward modification. The authors have also missed prior work with similar findings (e.g., LAST LAYER RE-TRAINING IS SUFFICIENT FOR ROBUSTNESS TO SPURIOUS CORRELATIONS, Kirichenko et al., 2023 and SURGICAL FINE-TUNING IMPROVES ADAPTATION TO DISTRIBUTION SHIFTS, Lee et al., 2023).
>
> Based on these points, I will maintain my scores as they are.

---

> > ### Author Response · Authors · 2024-11-26
> >
> > Thank you for the response! We would like to appreciate the opportunity to further discuss the points you raised.
> >
> > >
> >
> > ## 1. "Pruning weights can lead to unpredictable behaviors, which you have not empirically evaluated, e.g.  The Super Weight in Large Language Models"
> >
> > We respectfully disagree with the perspective on this critique for the following reasons:
> >
> > - Parameter pruning is a well-established practice in the context of LLMs [1]. Criticizing an entire family of methods based on the possibility that it may cause unpredictable behaviors seems speculative. Furthermore, while the reviewer references "The Super Weight in Large Language Models," our understanding is that this paper focuses on identifying specific parameters (as few as 0.01% of the model size) that significantly impact model behavior. Given that our method prunes 0.05% of the parameters, the likelihood of overlap between "The Super Weight" parameters and our identified "Bias Nodes" is low.
> >
> > - In addition, to demonstrate that pruning does not lead to significant impacts, we have presented model performance on two general NLP tasks above. If this does not provide sufficient empirical evidence, we kindly ask for clarification on what additional experiments would address your concerns and demonstrate that our Bias Node Pruning method does not cause unpredictable behaviors.
> >
> > [1] LLM-Pruner: On the Structural Pruning of Large Language Models
> >
> > >
> >
> > ## 2. "Altering an LLM to fix a single "niche" issue is, therefore, not convincing"
> >
> > We respectfully disagree with the characterization of Selection Bias as a "niche" issue. Below, we provide a list of works that focus exclusively on addressing Selection Bias, also referred to as Position Bias, Permutation Bias etc.:
> >
> > - J. Robinson et al., "Leveraging large language models for multiple choice question answering", ICLR 2023
> > - C. Zheng et al., "Large language models are not robust multiple choice selectors.", ICLR 2024
> > - A. Liusie et al., "Teacher-student training for debiasing: General permutation debiasing for large language models", ACL 2024
> > - P. Wang et al., "Large language models are not fair evaluators", ACL 2024
> > - X. Wang et al., "My answer is c”: First-token probabilities do not match text answers in instruction-tuned language models.", ACL 2024
> > - S. Wei et al., "Unveiling selection biases: Exploring order and token sensitivity in large language model", ACL 2024
> > - M. Xue et al., "Strengthened symbol binding makes large language models reliable multiple-choice selectors", ACL 2024
> > - P. Pezeshkpour and E. Hruschka, "Large language models sensitivity to the order of options in multiple-choice questions", NAACL 2024
> > - Y. Reif and Ro. Schwartz, "Beyond performance: Quantifying and mitigating label bias in llms.", NAACL 2024
> > - Z. Li et al, "Split and merge: Aligning position biases in large language model based evaluators", EMNLP 2024
> > - R. Li and Y. Gao, "Anchored answers: Unravelling positional bias in gpt-2’s multiple-choice questions", arxiv 2024
> >
> > Notably, the paper "Large language models are not robust multiple choice selectors" was recognized as a **Spotlight paper at ICLR 2024**, further highlighting the significance of this issue. These works demonstrate that Selection Bias is far from being a niche problem; it is a critical challenge that merits dedicated attention and research.
> >
> > >
> >
> > ## 3. "The authors have also missed prior work with similar findings"
> >
> > The two papers mentioned by the reviewer differ significantly from the focus of our work. Specifically, both papers explore fine-tuning methods to enhance certain model properties, whereas our approach is based on parameter pruning. We kindly request the reviewer to clarify the relevance of these works to our study, as this would help us better understand the connection being drawn.

---

### Meta-Review · Area_Chair_eS64 · 2024-12-19

**Metareview:**

The paper shows certain value in terms of innovation and experimental verification, but the deficiencies in experimental design and analysis, as well as writing and expression, have affected the quality of the paper, which suggests a reject.
The authors need to further improve the explanation of the experiments, optimize the description of the evaluation metrics, and enhance the clarity of writing to make the research more convincing.

**Additional Comments On Reviewer Discussion:**

The authors' responses have alleviated some of the reviewers' concerns to a certain extent, but there are still reviewers who think that the paper needs major revisions. The positive reviewers does not argue for accept.

---

### Decision · Program_Chairs · 2025-01-22

Reject